# Organ-specific, multimodal, wireless optoelectronics for high-throughput phenotyping of peripheral neural pathways

Woo Seok Kim [1], Sungcheol Hong[1], Milenka Gamero[1], Vivekanand Jeevakumar[2], Clay M. Smithhart[2], Theodore J. Price [2], Richard D. Palmiter[3✉], Carlos Campos[4✉] & Sung Il Park [1,5✉]

The vagus nerve supports diverse autonomic functions and behaviors important for health and survival. To understand how specific components of the vagus contribute to behaviors and long-term physiological effects, it is critical to modulate their activity with anatomical specificity in awake, freely behaving conditions using reliable methods. Here, we introduce an organ-specific scalable, multimodal, wireless optoelectronic device for precise and chronic optogenetic manipulations in vivo. When combined with an advanced, coil-antenna system and a multiplexing strategy for powering 8 individual homecages using a single RF transmitter, the proposed wireless telemetry enables low cost, high-throughput, and precise functional mapping of peripheral neural circuits, including long-term behavioral and physiological measurements. Deployment of these technologies reveals an unexpected role for stomach, non-stretch vagal sensory fibers in suppressing appetite and demonstrates the durability of the miniature wireless device inside harsh gastric conditions.

[1] Department of Electrical and Computer Engineering, Texas A&M University, College Station, TX, USA. [2] School of Behavioral and Brain Sciences, The University of Texas at Dallas, Richardson, TX, USA. [3] Department of Biochemistry, Howard Hughes Medical Institute, University of Washington, Seattle, WA, USA. [4] Division of Metabolism, Endocrinology and Nutrition, University of Washington, Seattle, WA, USA. [5] Institute for Neuroscience, Texas A&M University, College Station, TX, USA. ✉email: palmiter@uw.edu; camposca@uw.edu; sipark@tamu.edu

The nervous system consists of transcriptomically distinct neuronal cell types that reflect differences in sensing capabilities, connectivity, and function. Mapping their respective functions represents one of the major goals and challenges for modern neuroscience[1]. To this end, optogenetics has facilitated the untangling of neural networks by using light-activated, genetically encoded opsins to selectively manipulate the activity of distinct neuronal cell types with spatial precision[2]. However, this functionality has been limited to the brain due to constraints associated with peripheral light delivery: body tissues typically lack a stable interface for securing fiber optics and the inflexible nature of these optical probes would cause shearing of tissues and nerves during an animal's natural movements. As a result, a cell-type-specific understanding of the peripheral nervous system in freely behaving animals is severely lacking.

This is exemplified by the vagus nerve, which provides the only direct neural communication between internal organs and the brain. Peripheral endings of vagal afferent fibers respond to a broad array of stimuli, including hormones, osmolytes, changes in pH, and mechanical distention that have diverging functions and contributions to behavior[3]. All of its diverse sensory cell bodies reside together within the nodose ganglia[3], but conventional viral and transgenic methods for targeting genetically distinct neuronal populations do not permit organ-specific manipulations. Although pioneering studies have used fiber optics to optogenetically manipulate mouse vagal afferents with organ specificity, these studies were conducted under anesthesia to investigate autonomic functions[4,5]. Studying functions beyond reflexes, such as gastrointestinal mechanisms of satiation, requires a more flexible approach. Given the widespread interest in using vagal nerve stimulation for treating obesity and other neurological disorders[6,7], a key priority for this research field is to attain cell-type- and organ-specific manipulations of the vagus nerve in animals that are awake. Accordingly, we set out to develop a biocompatible, wireless optogenetic device for organ-specific light delivery.

Advances in wireless technologies have enabled the internalization of light sources, bypassing physical constraints associated with fiber-optic cables, and driving a shift in what is possible with optogenetics. Wireless, radio-frequency (RF)-powered devices were miniaturized for microscale light-emitting diode (μLED) insertion into the brain[8] and recently encased within stretchable and impermeable tethers for securing them onto subdermal tissues[9]. Despite these developments, organ-restricted illumination remains a challenge. A wirelessly powered μLED that is secured to the rat bladder using a circumferential elastomer sleeve could enable a similar level of functionality[10], but this approach impedes organ expansion. Efforts to wirelessly manipulate neural organ function in awake mice include studies that sutured an μLED onto the heart surface for pacemaking[11] or intestine surface for controlling colonic motility[12]. However, these devices were not described as being functional for >8 days[10–12], a limitation for conducting behavioral studies given the extended recovery periods required after thoracic and abdominal device implantation. Moreover, affixing the μLED to the target organ surface results in light back-scatter and non-specific optogenetic illumination of nearby tissues. No device has yet enabled chronic and durable cell-type-specific optogenetic manipulation of peripheral neurons inside of an organ.

Here, we describe the development of a durable, multimodal, wireless platform that enables optogenetic stimulation of peripheral neurons within organs in a high-performance manner. The miniaturized wireless device is fully implantable, utilizing a soft, thin, and low-modulus tether for targeting a μLED inside an organ. A unique fabrication method is employed to make a robust, μLED-housing tether, permitting long-term (>1 month), intimate interfacing with peripheral nerve endings in freely behaving mice. These optogenetic implants can selectively and independently manipulate peripheral nerve activity within multiple target organs in the same animal using a monolithic design. In addition, a channel isolation strategy is introduced for powering multiple cages using a single RF transmitter. Coupled with an advanced coil-antenna approach, a single telemetry system provides reliable wireless power in eight individual homecages, overcoming cage limitations of other wireless and fiber-optic-based systems. Precise targeting of a μLED within the stomach revealed an unexpected role for putative, gastric chemosensors in suppressing appetite and revealed a valence mechanism by which appetite suppression occurs.

## Results

**Organ-specific, wireless gastric optogenetic device.** An illustration of the fully implantable wireless device shows the general strategy for targeting a μLED inside the stomach (Fig. 1a, b). The device consists of an analog, front-end electronic circuit for RF harvesting (5.5 mm radius and 1 mm thickness) and a tether that supplies current to a μLED. It harvests RF energy from a remotely located wireless RF-power system, converts RF energy into optical energy, and illuminates targeted regions in the stomach. The μLED is situated in the middle rather than the end of a tether, allowing the tether to be threaded in and out of the stomach and secured at two contact points. We found that the tether remains secure with purse-string sutures. The ultra-thin tether (0.4 mm wide by 0.2 mm thick) is more than three times smaller than insulin syringe needles used for intraperitoneal injections and tubing used for intragastric infusions[13].

Essential features that allow for long-lasting operation of the ultra-thin tether are a pre-curved, sandwiched construction. In our prototype, the harvester and μLED were connected with thin copper (12 μm) electrical interconnects on top of flexible and durable polyimide (18 μm) substrate, and then coated with a biocompatible silicone polymer, polydimethylsiloxane (PDMS). However, this design exhibited poor durability and post hoc analysis revealed μLED tether damage likely caused by a mechanical strain. To increase durability, the μLED was sandwiched in between a second copper/polyimide bilayer, which also provided additional electrical contact (Fig. 1c; assembly steps 1 and 2). We further postulated that coating the tether with silicone in a curved position (pre-curved) would decrease strain compared to a tether that was coated in a flat orientation and then bent when securing it inside the stomach (post-curved). This was achieved by suspending the tether in a bent position, pipetting small amounts of melted silicone around the μLED, and coating the remaining components using a simple dipping process (Fig. 1c; assembly step 3). This resulted in a thin, soft, and lightweight (~380 mg), wireless, gastric optogenetic implant (Fig. 1c; assembly step 4). The compliant, low-modulus properties eliminated constraints on the natural motions of the animal while also minimizing mechanical strain at the connecting joints.

Three-dimensional (3D) modeling of the mechanics showed that the maximum strain in the copper traces and PDMS coating of the pre-curved tether (<800 Pa) was dramatically reduced compared to the post-curved tether strain (<6600 Pa) (Fig. 1d and Supplementary Fig. 2). We further optimized the tether by mechanically testing various curvatures and identified a pre-curved configuration that was functional beyond 200 kilocycles (kc) (Fig. 1e and Supplementary Fig. 3). Lifetime mechanical cycle tests with a significant load (0.03 kgF) revealed that the pre-curved structure with a radius of 1.15 mm was functional for 200 kc, a nearly 10-fold improvement compared to the post-curved structure (Fig. 1f and Supplementary Fig. 4). Although

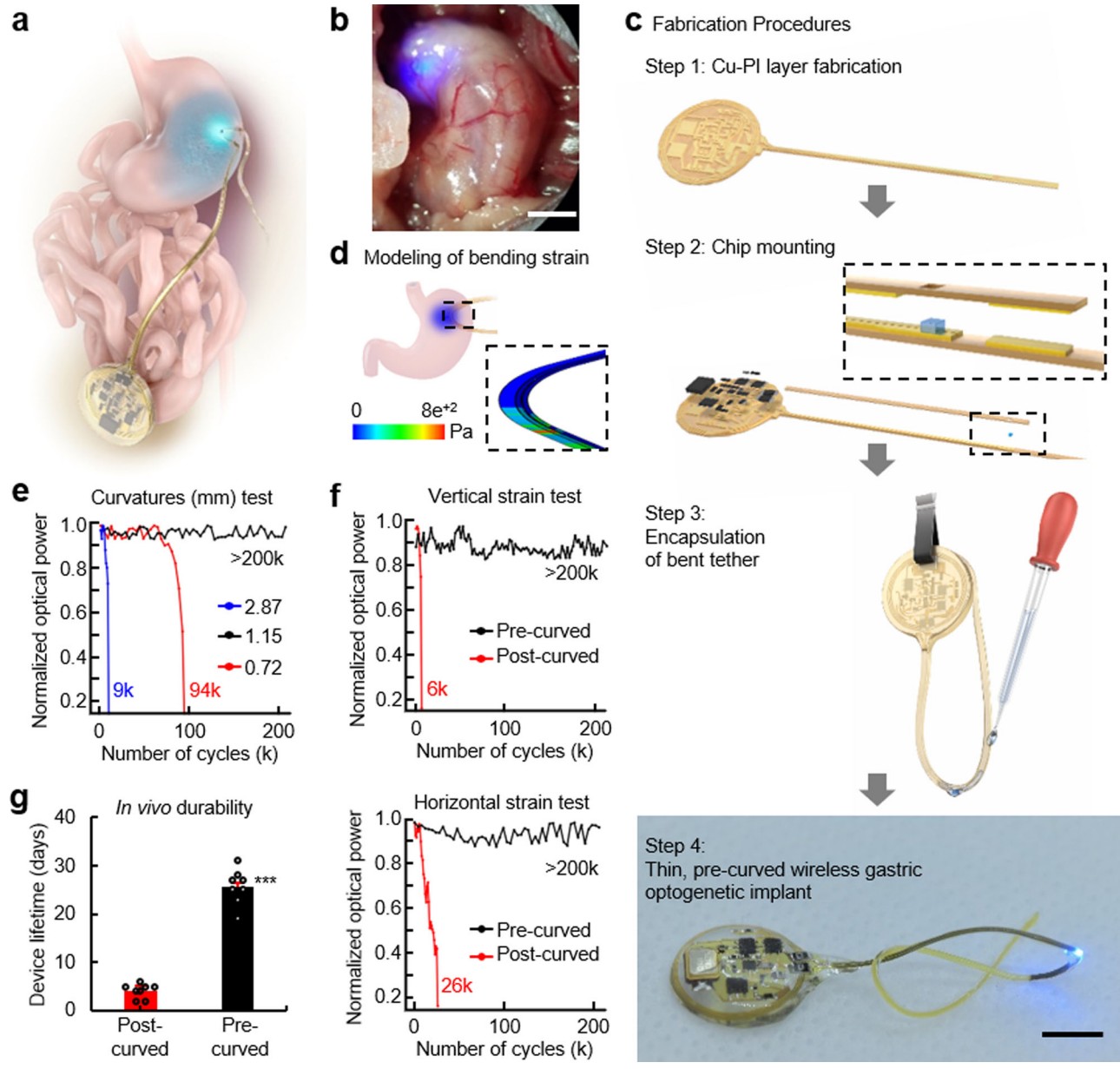

**Fig. 1 Development of a soft, wireless gastric optogenetic implant with a pre-curved, sandwiched tether. a** Illustration of a soft, wireless gastric optogenetic implant with a pre-curved, sandwiched tether. **b** Image of wireless LED operation in the stomach; scale bar 5 mm. **c** Procedures for device fabrication; scale bar 5 mm. **d** Three-dimensional modeling of the mechanics for the pre-curved, sandwiched structure. **e** Plots of output power vs. curvatures of a tether. The legend numbers represent the curvature of the device as the length of the arc for a radius of 2 mm (see Supplementary Fig. 3 for equation); smaller numbers represent sharper U-shape angle. **f** Measurement results of device lifetime cycling test for both structures when strain applied in the horizontal (top) and vertical direction (bottom). **g** Measurements of device lifetime for the pre- and post-curved structure when implanted (pre-curved, $n = 8$; post-curved, $n = 8$). Bar graphs are mean ± SEM. Statistical comparison was made using two-tailed $t$ test; ***$p < 0.001$.

there was improved durability with pre-curved structures that had a radius of 2.87 and 0.72 mm, they were not as durable as 1.15 mm, likely because 2.87 mm is too similar to the flat structure, whereas the sharp angle with a radius of 0.72 mm interferes with μLED contact with the pad. The device was also subjected to waterproof testing by submerging into a heated saline solution, revealing that it remained continually functional for over 2 months, even in extreme temperatures (Supplementary Fig. 5). Heat dissipation is another factor that can limit device functionality since nerve endings in the gastrointestinal tract can be temperature sensitive[14]. Thermal assessment of the wireless optogenetic implant demonstrated minimal temperature increases (~0.2 °C; Supplementary Fig. 6) during typical operating

conditions (10 and 20 Hz with 5 ms light pulse; 10 and 5% duty cycles). Consistent with this, calculation of specific absorption rate (SAR) using a finite-element method analysis tool showed that the SAR distribution against localized RF exposure is below IEEE guidelines[15] (Supplementary Fig. 7). Finally, tests in mice showed that the pre-curved, sandwiched tether was functional for over a month, while the post-curved structure stopped working 3 days after implantation (Fig. 1g).

**Efficient and high-throughput, wireless optoelectronic systems.** The practical use of optogenetics depends on reliable and cost-effective light delivery in multiple animal subjects. A complete

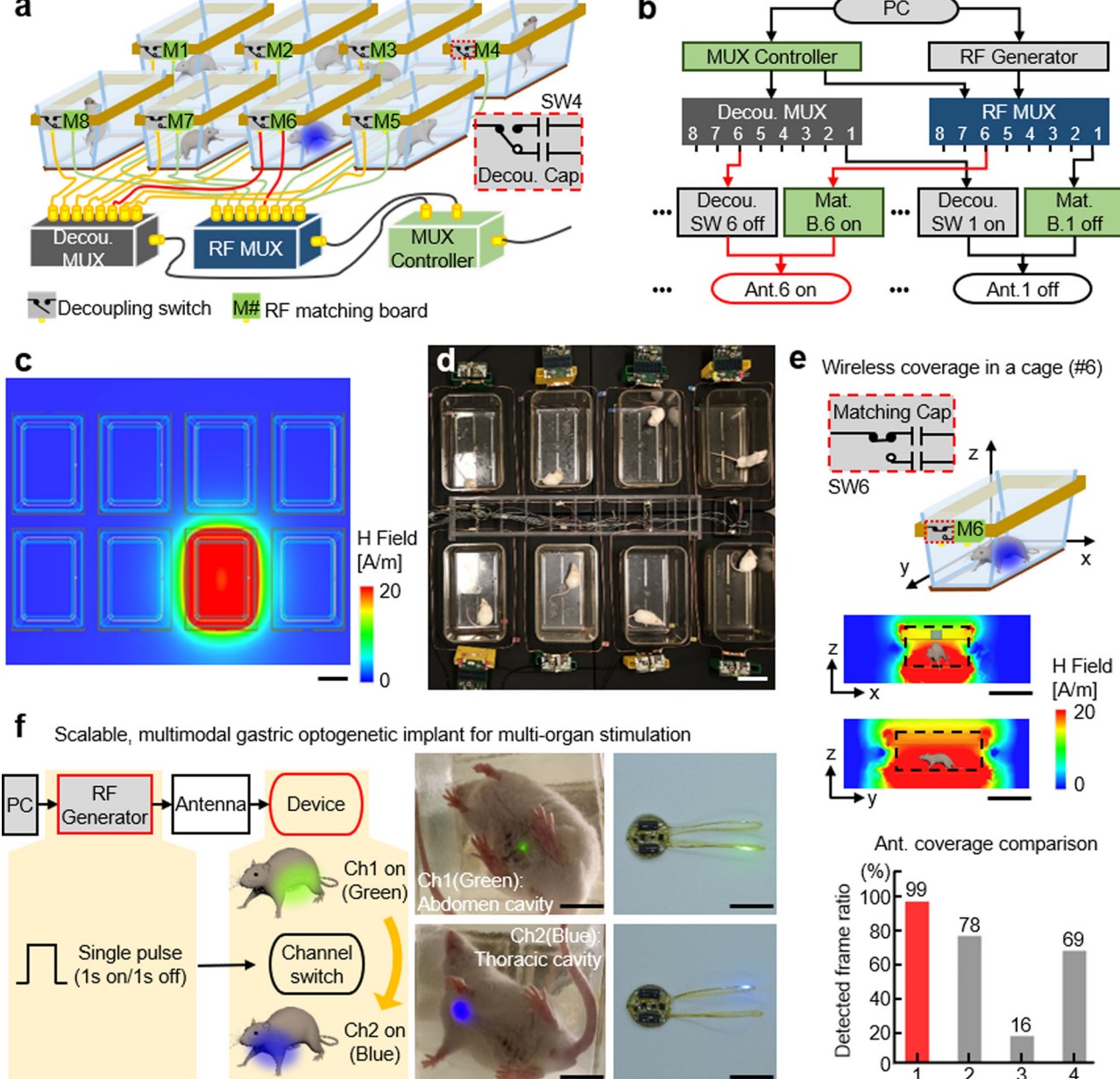

**Fig. 2 Electrical characteristics of multiple cage wireless power TX system and multiple organ stimulation devices. a** Schematic illustration of the proposed wireless power TX system for high-throughput phenotyping of neural pathways. **b** Functional block diagram of the proposed wireless power TX system. **c** Electromagnetic simulation of wireless coverage for the proposed TX system; scale bar 10 cm. **d** Picture of the TX system; scale bar 10 cm. **e** Representative magnetic field distributions in antenna set 6 (top) and comparisons of wireless coverage for the proposed system and other wireless power TX systems (bottom); scale bar 10 cm. **f** Illustration of wireless operation of a scalable, multimodal wireless gastric optogenetic implant (left), images of an animal with the device implanted (middle), and image of the device (right); scale bar 1 cm. MUX; multiplexer, Decou.; decoupling, SW; switch, Cap; capacitor, Mat. B.; RF matching board, Ant.; antenna set.

laser-based, optogenetics setup remains cost-prohibitive for many labs, given that each animal subject requires a laser, fiber-optic cannula, fiber-optic patch cord, and rotary joint to decrease physical constraints of a patch cord[16]. Wireless optogenetics is similarly limited, typically requiring a single RF-power generator for each homecage[17]. Multiple RF-power generators can be used, but they must be operated at least 1 m apart from each other to avoid electromagnetic interference[18] (Supplementary Fig. 8). Together, these constraints limit the group sizes used for studies, restrict the duration and type of behavioral experiments that can be conducted, and overall prohibit the high-throughput utilization of optogenetics.

To overcome these limitations, we developed a multiplex approach to power eight individual cages with a single RF-power generator. The wireless telemetry system consists of an RF-power generator, controller, RF multiplexer, decoupling multiplexer, and an antenna set for each of the 8 cages (Fig. 2a); each antenna set is made of a pair of top and bottom coil structure. Simultaneous and independent control of the eight cages is achieved with coupling and decoupling circuits that manage the tuning of antennas to operational (13.56 MHz) and non-operational (100 MHz) device powering frequencies. For example, when the controller selects antenna set 6, the RF multiplexer tunes antenna set 6 to 13.56 MHz and decoupling multiplexer detunes the other antenna sets

to 100 MHz (Fig. 2b). The other antennas that are detuned and deactivated can only pass a negligible amount of energy at a frequency of 100 MHz, which significantly deviates from the resonant frequency of 13.56 MHz. Therefore, the other seven antenna sets do not cause interference even when directly adjacent to the actuating 13.56 MHz antenna; this was confirmed with electromagnetic simulation results and validation experiments in vivo (Fig. 2c, d, Supplementary Movies 1, 2, and Supplementary Fig. 9). Since optogenetics typically requires brief intermittent light pulses to avoid depolarization block[16], this strategy can be used to toggle between multiple cages to deliver intermittent light pulses. Therefore, the limiting factors for the number of cages that can be operated simultaneously with a single RF-power generator are the stimulation frequency and duration of light pulses. With the proposed arrangement (Fig. 2a), we conducted experiments within eight cages simultaneously using 20 Hz and 5 ms pulse duration stimulation parameters. This extends the high-throughput utilizing optogenetics to do the experiments with at least eight mice (supposed a mouse in a cage), a typical group size, at that same time. For example, measurements of food intake require 4 h for each animal in the group. To complete the analysis of feeding behavior for two groups of animals (experimental and control), each of which has eight animals, it only takes 8 h, while approaches using existing wireless TX system, a single power source coupled with a single cage, demands 64 h ($8 \times 8$ h). This makes it less ideal for longitudinal experiments, in particular those required for most obesity experimental designs, where a device needs to be chronically implanted for >2 months. Also, through the modification of the multiplexer board and controller, simultaneous activation of 16 cages at 10 Hz or 32 cages at 5 Hz frequency is possible when using a 5 ms light pulse duration.

In addition to couplings, wireless coverage remains a significant limitation for optogenetic experiments. Conventional systems utilize a single RF antenna below or around the sides of a homecage[17,19]. Due to electromagnetic dissipation away from the RF source, wireless coverage can be as low as 30% in a homecage[20] and worse in larger behavior boxes. Previously, these limitations were circumvented by increasing RF power, but this results in undesired RF energy to animal tissues and increased heat generation. A recent study utilized an RF multiplexer that rapidly toggled power between two antennas to increase coverage[21]. However, this approach requires careful operation and validation to avoid electromagnetic interference between the antennas, and limits the use of a multiplexer for powering multiple cages. Here, we introduce a simple dual-coil-antenna system for increasing wireless coverage[22]. It consists of a top antenna coil that is connected to an RF generator and an unconnected antenna coil below the cage that passively attracts RF signals towards the animal subject and cage bottom. Three-dimensional electromagnetic modeling suggested that the dual-coil-antenna system could enable continuous operation throughout a location of interest (Fig. 2e; top and Supplementary Fig. 10). This was confirmed with light-power-output measurements of wireless devices at five representative positions and heights from the cage bottom, which demonstrated robust device activation throughout the volume of a cage (Supplementary Fig. 11). Furthermore, the dual-coil-antenna system eliminated the dependence of transmitted power on the relative orientation angle between the transmission antenna and the device (Supplementary Fig. 12). Comparison studies further indicated that the proposed antenna system outperforms other existing systems, offering virtually complete wireless coverage in a homecage (Fig. 2e (bottom) and Supplementary Fig. 13).

**Scalable, multimodal device operation**. Multimodal device operation is another strategy for increasing the efficiency and throughput of wireless optogenetic studies. Targeting multiple organs with a single device could enable multiorgan analysis in the same animal or even be used to examine organ-to-organ interactions. Realization of multimodal tools requires an actuation mechanism that can remotely manage channel selection. Previous efforts utilized higher operating frequencies, microcontroller chip, or Bluetooth kits for actuating separate channels, but these approaches require increased RF power (tens of mW) for operation and render them energy-hungry devices[10,21]. Here, we use a reed switch in the device that responds to the pattern of externally applied electromagnetic RF pulses. In our example, a pulse width longer than 100 ms triggers the transition from a green μLED to a blue μLED located on a separate tether (Fig. 2f and Supplementary Movie 3). The actuation threshold can be adjusted by pairing different capacitors and resistors with the reed switch to prevent unwanted activation or deactivation, and could theoretically be tuned for switching between more than two channels[23,24]; the circuit diagram is shown in Supplementary Fig. 14. Importantly, this strategy only requires 10 μW for channel selection, which is 100-fold less power than other approaches[10,21,25]. When combined with the dual-coil antenna and multiplex coupling/decoupling, the proposed optoelectronic system enables robust, ultra-efficient wireless powering of optogenetic devices in multiple organs and multiple cages with independent and simultaneous control. Time slots allocated for each cage and threshold pulse for channel selection are tunable, suggesting many scenarios of multiplexing and multimodal operation. For example, we set the threshold pulse for activation/deactivation of channels to 100 ms and allocate 250 ms for each cage. This provides enough time for an implant in each cage to switch its channel (from Ch1 to Ch2 or vice versa). It requires only 2 s ($8 \times 0.25$ s) for the switching operation of implants in cages. Next, the TX system can adjust time slots, depending on stimulation conditions.

**Optogenetic manipulation of gastric vagal sensory endings**. To determine the utility of the optoelectronic system, we investigated the role of stomach vagal afferent endings in feeding behavior. We began by analyzing the μLED light spread and identified RF powering parameters needed for organ specificity. As expected, securing the μLED inside the stomach significantly restricts light spread, in contrast to surface affixation, which results in light back-scatter intensities well above the threshold for opsin activation[26] (Fig. 3a and Supplementary Fig. 15). We examined whether implantation of the stomach device is well tolerated by showing that ad libitum food intake of mice implanted with the device was the same as sham-operated mice (Fig. 3b). These results indicate that our wireless device should allow precise optogenetic manipulations in awake, behaving mice.

A recent study identified genetically distinct vagal afferent neurons in the nodose ganglion that innervate the stomach and express either *Calca*, *Sst*, *Gpr65*, or *Glp1r* genes[27]. In contrast to *Sst* and *Gpr65*, which exhibit either mechanosensitive morphological endings in muscle layers (*Glp1r*) or chemosensitive endings in the mucosal layer (*Sst* and *Gpr65*), *Calca*+ neurons form spatially restricted chemosensitive mucosal endings in the corpus vs. mechanosensitive intramuscular arrays in the stomach antrum[27]. Identification of a role for stomach chemosensation in appetite control has been elusive[28]; therefore, we used our wireless device to selectively activate *Calca*+ vagal afferent chemosensitive endings in the corpus region of the stomach.

To gain cell-type specificity, AAV9 was injected into the nodose ganglion of *Calca^Cre* transgenic mice to introduce

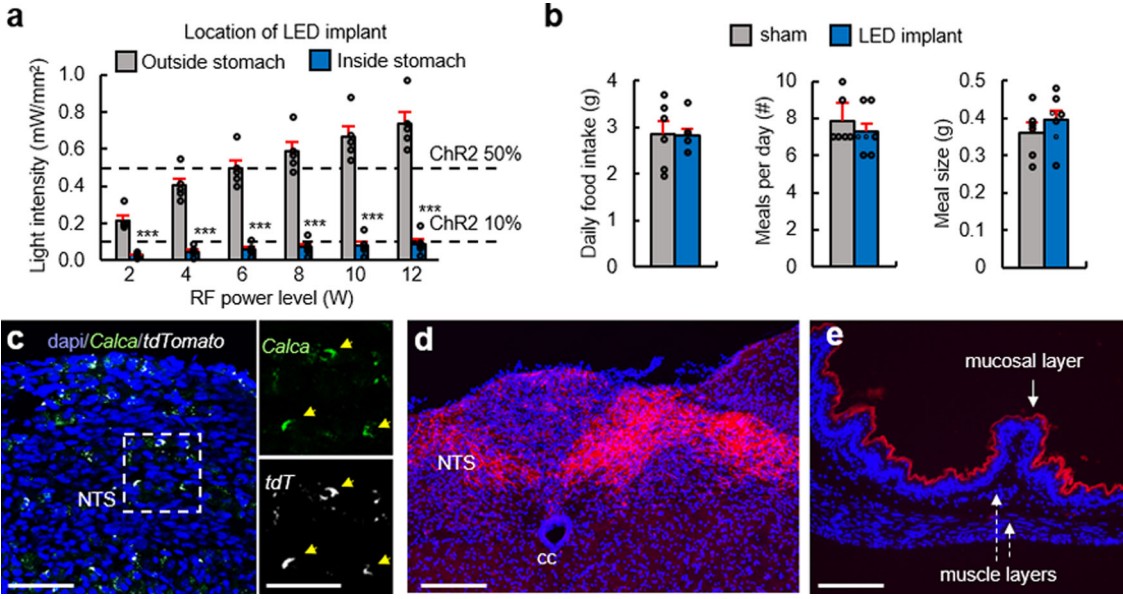

**Fig. 3 Optogenetic targeting of *Calca*+ vagal afferents in the stomach. a** Light intensity measurements comparing LED implantation inside vs. outside the stomach ($n = 5$, $p < 0.01$), with varying RF powers ($p < 0.001$). Dashed horizontal lines indicate light intensity needed for 10 and 50% maximal activation of channelrhodopsin2. **b** Comparison of total food intake, number of meals, and meal size in mice implanted with LED device ($n = 7$) or sham operated ($n = 6$) ($p = 0.71$). **c** *Calca^Cre* transgenic mice received nodose ganglion injection of AAV9-DIO-ChR2:tdTomato. Images show fluorescence in situ hybridization of tdTomato and Calca mRNA, demonstrating the cell-type specificity of transgenic/viral approach; scale bars 25 µm. **d** tdTomato fluorescence labeling of central *Calca*+ vagal afferent endings in the nucleus of the solitary tract (NTS); scale bar 25 µm. **e** Fluorescence labeling of peripheral *Calca*+ vagal afferent endings in the stomach mucosal layer; scale bar 50 µm. Behavioral experimental results are from one cohort of animals. Bar graphs are mean ± SEM. Statistical comparisons were made using two-way repeated-measures ANOVA, Tukey's post hoc; \*\*\*$p < 0.001$.

Cre-dependent ChR2:tdTomato opsin expression or a control group with just tdTomato fluorescent reporter (Fig. 3c). Precise anatomical specificity was achieved by implanting the µLED into the fundus, immediately adjacent to the corpus (Fig. 3d). While *Calca*+ vagal afferents do not innervate the fundus, we implanted the device away from the antrum to avoid activation of mechanosensitive fibers (Fig. 3e). Several weeks after recovering from device implantation, mice were fasted overnight and refed the following morning. Compared to no stimulation (RF antenna off), optogenetic activation produced robust suppression of food intake during refeeding, with greater stimulation frequencies almost completely suppressing intake (Fig. 4a, b and Supplementary Movie 4 (left)). Importantly, activation of the device in control group without ChR2 did not alter feeding behavior, indicating that RF signals and activation of the device in itself do not influence feeding (Fig. 4c and Supplementary Movie 4 (right)). To further establish that the appetite suppression was due to activation of vagal afferent endings in the stomach, we compared these results to separate cohorts of mice implanted with six non-attached µLEDs in the abdomen. Although optogenetic stimulation of vagal afferents in this manner suppressed feeding, the effect was not as robust despite stimulating with six µLEDs rather than a single µLED directly implanted inside the stomach (Supplementary Fig. 16). Furthermore, the non-anchored LED approach required increased operating power to compensate for light dissipation, resulting in greater heat generation and potential tissue damage[29,30]. Thus, the gastric optogenetic implant enables more robust optogenetic activation using less wireless power.

Appetite suppression can be associated with positive valence, potentially due to the removal of aversive hunger signals[31], or aversion in response to harmful stimuli, such as uncomfortable gastric distension[32] or food poisoning[33]. To investigate the

affective mechanisms by which *Calca*+ gastric vagal afferent neurons might suppress appetite, we constructed oversized dual-coil antennas for robust optogenetic activation in various behavior boxes (Fig. 4d, g and Supplementary Fig. 17). In one assay, mice were placed in a two-chamber box with RF power only in one chamber to determine whether mice form an aversion or preference for the optogenetic stimulation chamber. Surprisingly, we did not observe differences in place preference or avoidance (Fig. 4e, f), similar to optogenetic stimulation of other vagal afferent cell types that innervate the gastrointestinal tract[27]. Conversely, an open-field assay demonstrated that optogenetic stimulation reduced the time mice spent in the center, indicative of anxiety-like behavior and suggesting that activation of *Calca*+ gastric vagal afferent fibers might be aversive (Fig. 4h, i). We analyzed the locomotor activity from open-field and place-preference tests, which revealed decreased locomotion during optogenetic activation of *Calca*+ gastric vagal afferent (Supplementary Fig. 18); presumably, the mice feel aversion and have decreased motivation. In addition, gastrointestinal signals are closely associated with taste-sensory signals[34], suggesting that a learned food preference/aversion assay might be most indicative of the mechanism underlying appetite suppression. To test this hypothesis, mice were habituated to overnight water restriction for several days and then given access to 5% sucrose solution, followed by optogenetic stimulation of *Calca*+ vagal afferent fibers for 4 h. Three days later, mice were offered the choice of water or 5% sucrose. This two-bottle preference test revealed that activation of stomach *Calca*+ vagal afferents conditioned mice to avoid the sucrose solution (Fig. 4j). This suggests that appetite suppression occurs via a negative valence mechanism that alters taste preferences. These results identified a role for stomach mucosal *Calca*+ vagal afferents in appetite suppression and revealed a mechanism by which appetite suppression occurs.

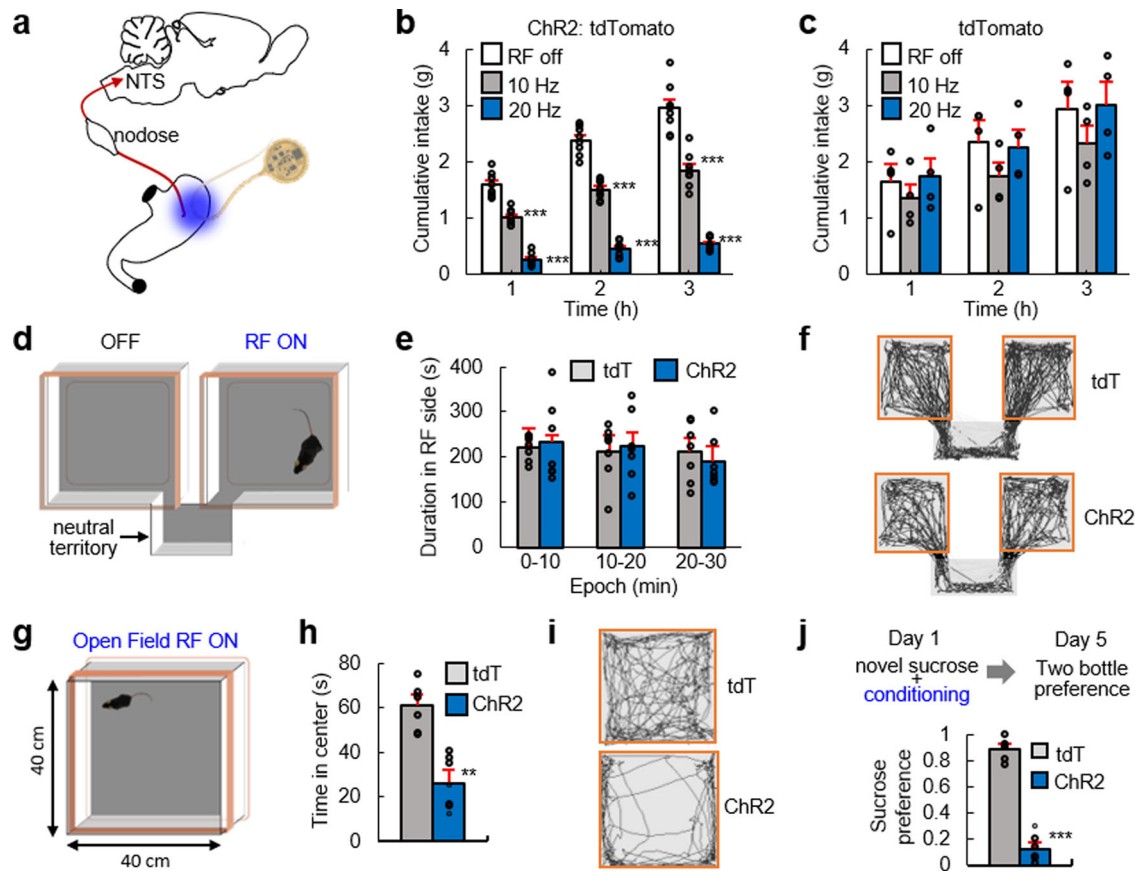

**Fig. 4 Activation of *Calca*+ stomach vagal afferents suppress appetite via negative valence mechanism. a** *Calca*$^{Cre}$ transgenic mice received a left nodose ganglion injection of AAV9-DIO-ChR2:tdTomato or AAV9-DIO-tdTomato control virus. The LED was implanted in the stomach corpus-function junction. **b** Frequency-dependent suppression of food intake in the ChR2:tdTomato group ($n = 8$). **c** The tdTomato control group did not suppress food intake during photostimulation ($n = 4$) ($p = 0.06$). **d** Illustration of real-time place-preference (RTPP) box. The RF antenna powered the device only in the right chamber. **e** Activation of LED device (20 Hz light pulses) did not induce a place preference nor avoidance in both ChR2 and tdTomato groups ($n = 7$ per group) ($p = 0.31$). **f** Representative traces for RTPP assay. **g** Illustration of a large open-field box; the antenna delivered wireless power throughout the entire arena (20 Hz light pulses). **h** Photoactivation of *Calca*+ gastric vagal afferents decreased time spent in center ($n = 7$ per group). **i** Representative traces from open-field test. **j** Mice were exposed to a novel sucrose solution on Day 1 followed by optogenetic activation of vagal sensory fibers (20 Hz). On Day 5, mice were water-restricted overnight and then given simultaneous access to a bottle of sucrose and a bottle of water. The graph is the sucrose preference score (ChR2, $n = 7$; tdT, $n = 5$). Experimental results are from one cohort of animals. Bar graphs are mean ± SEM. Statistical comparisons were made using two-way repeated-measures ANOVA, Tukey's post hoc, except for (**h**) and (**j**), which were two-tailed *t* test; ** $p < 0.01$, *** $p < 0.001$.

## Discussion

While many classical studies have established an important role for visceral signals in controlling behavior[35,36], these surgical- and chemical-denervation experiments lacked organ specificity and did not reveal the identity of sensory neurons that can serve diverging functions. Here, we developed wireless μLED devices that permit organ-specific, optogenetic manipulations, and an ultra-efficient wireless telemetry system for powering multiple cages. The miniaturized wireless device enabled precise optogenetic stimulation of genetically defined vagal afferents innervating the mouse stomach, revealing a function for *Calca*+ mucosal sensory endings in suppressing food intake via a negative valence mechanism. Critically, the pre-curved sandwich construction significantly extended the lifespan of the μLED device and allowed for testing various stimulation parameters and behavior tests within the same subjects. We envision that the current device could be used to optogenetically manipulate neural circuits throughout the gastrointestinal tract and other hollow organs, such as the bladder with little or no modifications.

Prior methods for optogenetic activation of vagal afferents in awake mice have either lacked organ specificity[27] or involved gut injections of a retrogradely transported opsin virus and fiber-

optic implantation in the hindbrain where vagal afferents terminate[4]. While the latter provides organ specificity, retrograde viruses can be limited by tropism and incomplete infection of certain cell types. Moreover, vagal afferents express neuropeptides in their peripheral endings which are hypothesized to be released in the gut to exert efferent functions[7]. In other systems, such as for somatosensation, the peripheral release of neuropeptides by afferent fibers can contribute to behaviors by sensitizing other afferent subtypes to ongoing stimuli[37]. Finally, fiber optics cannot be used for optogenetically manipulating the enteric nervous system nor splanchnic sensory afferents, which synapse in the spinal cord. Investigating the function of these neural circuits and hypotheses, therefore, requires peripheral optogenetic stimulation that is now possible with the proposed wireless gastric optogenetic implant. Multimodal features could further enable the investigation of peripheral interactions by using different colored μLEDs to activate corresponding color-sensitive opsins expressed by separate neural substrates or multiple organs simultaneously/independently.

In addition to extending optogenetic functionality to the peripheral nervous system, we introduced advancements in wireless telemetry that generally improve the scalability and usability of

optogenetics. The dual-coil-antenna system, which enables reliable and complete wireless coverage, is easily constructed using inexpensive copper wire secured onto cardboard or plastic backing. The multiplexing approach further allows for the testing of large experimental cohorts, which was particularly important for our studies because of the extended duration of feeding behavior tests. This system can be set up in under an hour, is simple to operate, and dramatically decreases the cost and time required for conducting optogenetic experiments. Furthermore, the wireless telemetry system has broad applicability for powering optogenetic devices in the periphery, brain[9,38], or other wireless devices, such as those that measure bioelectrical signals[10].

Future studies may take advantage of these enabled wireless optoelectronic features to chronically activate neural circuits for days, weeks, or even months. Because adaptations can occur with sustained activation of a neural pathway, such experiments are important for investigating the persistence of long-term, physiological effects, including weight loss. This can enable experiments that determine whether appetite suppression induced by gastric vagal afferent activation is attenuated in obese mice and whether chronic activation of vagal afferent endings in the stomach can reverse obesity. Identification of viscerosensory pathways that can either suppress or stimulate appetite will have direct clinical importance for potentially developing novel therapeutic targets for treating appetite disorders.

## Methods

**Device fabrication**. The process began with flexible copper/polyimide (Cu/PI) bilayer films (thickness; 12 μm/18 μm, AC181200RY, Dupont™ Pyralux®) mounted onto a glass slide (dimensions, 5.08 cm × 7.62 cm). Then, we deposited 2.5-μm thickness of photoresistor on the Cu/PI substrate (AZ 1518, AZ®, recipe; spin coated at 3000 r.p.m. for 30 s), and used UV photo-lithography to define patterns for pads and interconnections (EVG610, EV Group, recipe; UV intensity for 200 mJ cm⁻²). This was followed by immersion in developer solution (AZ Developer 1:1, AZ®) for 30 s and rinses in distilled water for 10 s. Immersion in copper etchant (LOT: Z03E099, Alfa Aesar™) for 7 min and rinses with acetone, methanol, isopropanol, and distilled water for 1 min yielded Cu interconnections and pads on the flexible substrate (Fig. 1c; step 1). After samples dry, chip components were mounted, including a μLED, passive components, and IC components using a soldering machine. An additional PI/Cu layer (18 μm/12 μm thick) with the bottom chip-mounted Cu/PI substrate formed a sandwiched structure (PI/Cu/Cu/PI) (Fig. 1c; step 2). For encapsulations, we applied a small amount of PDMS (Sylgard™ 184 Silicone Elastomer Kit, Dow®; 10:1 mix ratio) using a pipette while a clamp held the body of a sample to form a thin, pre-curved, sandwiched structure (Fig. 1c; step 3). Then, we encapsulated the body of a sample with PDMS by a dip-coating process (500 μm thick). Samples were cured in a vacuum oven at 100 °C for 1 h. These procedures yield a soft, low-power, wireless gastric optogenetic implant with a pre-curved, sandwiched tether (Fig. 1c; step 4). Detailed information on device layouts, IC components, and procedures for fabrication are found in Supplementary Fig. 1 and Supplementary Table 1.

**Finite-element method analysis**. For numerical electromagnetic simulations of the proposed device, we used a finite-element method analysis tool (Ansys Electromagnetics Suite 17-HFSS, Ansys®) with Cole-Cole dielectric relaxation model where characteristics of biological tissues were described as a function of frequency. Organ systems and tissues of a mouse were modeled to one million meshes for numerical simulations, and antenna coils made of copper stripes or wires were modeled to materials with finite conductivity, 58 MS s⁻¹. For three-dimension modeling of the mechanics for the devices, we used a commercial finite-element method analysis tool (Abaqus/CAE 2018, Dassault Systems) to investigate strain effects on the pre-curved and post-curved structures. The following parameters were used for simulations: thickness 500/18/12/12/18/500 μm (PDMS/PI/Cu/Cu/PI/PDMS) for the pre-curved structure and 510/12/18/510 μm (PDMS/Cu/PI/PDMS) for the post-curved structure; elastic properties Young's modulus (MPa)/Poisson's ratio: 1/0.49 for PDMS, 119000/0.34 for Cu, and 2500/0.34 for PI. Cu/PI layer was modeled as a composite shell element (S4R). PDMS was modeled as a solid hexahedron element (C3D8R) in the pre-curved structure and as a shell element (S4R) in the post-curved one. The mechanical simulation results are found in Supplementary Fig. 2.

**Mechanical, optical, and electrical measurements**. We used a gauge-force machine (ESM303 Forced Test Stand, MARK-10) to perform device lifetime cycling tests with a significant load extended over a period of time (>200 kc) for the pre-curved, post-curved, and pre-curved structures with three different curvatures

(0.72, 1.15, and 2.87 mm) of a tether. The experiments involved the application of strain in three different directions: (1) the x-direction, (2) the y-direction, and (3) the z-direction, respectively. After completion of each 1000 cycles, we immersed a wireless device in 10% phosphate-buffered saline (PBS) solution for 10 min and measured light intensity using a light meter (LT300, Extech). This test was repeated until a device stopped functioning (Fig. 1e, f and Supplementary Fig. 4). We also performed accelerated life testing where a device was immersed in 10% PBS solution and light intensity was monitored as a function of time at various temperatures (25, 60, and 90 °C) (Supplementary Fig. 5). For thermal assessments of wireless devices, we used an infrared camera (VarioCAM HDx head 600, InfraTech). Light intensity was fixed at an optical intensity of 10 mW mm⁻², which is enough to activate light-sensitive proteins, and the camera measured variations in temperature when devices were operated with duty cycles of 20, 40, 60, 80, and 100% (Supplementary Fig. 6).

**Antenna-coil fabrication and wireless, power-control system**. We used 8-gauge bare Cu wire for the bottom antenna coil and Cu stripes (0.635 mm thick by 2.54 cm wide) for the top antenna coil. The bottom coil was placed under a cage while the top coil was situated 8 cm above the cage bottom. Impedance matching using Network Analyzer (ENA Series E5063A, Keysight) with a discrete capacitor component yielded two antenna coils, each of which resonates at 13.56 MHz (the top coil) and 15 MHz (the bottom coil), respectively; these different frequencies offer broad bandwidth and stable coverage. Wireless power-control systems consisted of an RF-power supply (ID ISC.LRM2500-A, FEIG Electronics), matching board (ID ISC.DAT-A, FEIG Electronics), RF multiplexer (ID ISC.ANT.MUX.M8, FEIG Electronics), controller (nRF52832 Development Kit, Nordic semiconductor), and decoupling multiplexer. The controller was programmed a custom C code based on C code libraries from Nordic (nRF5_SDK_13.0.0_04a0bfd) by Keil uVision 5 IDE (μVision V5.23.0.0). The circuit diagram, device layout, and information on decoupling multiplexer are found in Supplementary Fig. 9. Dimensions of a representable coil antenna, capacitance, and inductance for different sizes of experimental assays including homecages are found in Supplementary Table 2.

**Measurements of wireless coverage**. We implanted a wireless device over the skull, under the skin of a mouse, and recorded their behaviors using three cameras (C615, Logitech). A red-colored μLED was embedded in an implanted device to serve as a signal that can be easily detected by cameras over a cage and the wireless TX system transmitted RF signals at 1 W. One camera was positioned above a cage and two cameras recorded from left and right sides. They recorded behaviors of an animal in a cage for 2 min and we extracted images from the recordings and analyzed them frame by frame to determine whether an image had captured wireless operation of a device (red μLED). Next, we counted the number of frames missing wireless operation. For the purpose of visual demonstration of wireless coverage, we reconstructed 3D continuous traces of a red μLED from extracted images using custom scripts in Python (version 3.7.3–64 bit, Spyder 3.3.6 IDE). We repeated the procedures described above for other wireless antenna technologies (Supplementary Fig. 10). For validations of wireless power TX systems, we used an electromagnetic probe (TBPS01-TBWA2/40 dB, Tekbox) to measure the output power at five representative positions (A, B, C, D, and E) and various heights from the bottom of the enclosure as a function of the distance and angle (Supplementary Figs. 11 and 12).

**Mice**. Calca^{Cre:GFP} homozygous mice (C57Bl/6 background; Jackson laboratory Calca^{tm1.1(cre/EGFP)Rpa}) were bred with C57Bl/6 mice to generate Calca^{Cre:GFP/+} heterozygous mice used in experiments[39]. Following surgery, mice were singly housed with ad libitum access to standard chow diet (LabDiet 5053) in temperature- and humidity-controlled facilities with 12 h light/dark cycles. Both male and female mice were used for behavioral experiments. All animal care and experimental procedures were approved by the Institutional Animal Care and Use Committee at the University of Washington.

**Organ-specific, wireless, gastric optogenetic device implantation**. Under surgical anesthesia (isoflurane, 1–2% inhalation), the animal's ventral side was shaved, sterilized with three alternating scrubs of betadine and alcohol, and the surgical field was restricted with sterile drapes. With the animal on its back, a 2 cm skin incision was made along the abdominal midline from the xiphoid cartilage extending to the mid-abdomen, and a second cut into the abdominal wall exposed the stomach for device implantation. Ringed forceps were used to gently grasp the fore-stomach and pull it out of the abdominal cavity onto gauze soaked with sterile saline. Fine-tipped Dupont forceps were then used to puncture the stomach fundus and thread the μLED tether in and out of the stomach. With the μLED inside the stomach, the tether was secured in place with purse-string sutures (5-0 PGA). The device harvester was then placed in the abdominal cavity and the stomach was placed back into its normal orientation. The abdominal wall was closed with interrupted stitches using absorbable suture (5-0 PGA), and the skin with non-absorbable suture (6-0 silk). Mice received analgesics during the surgery (ketoprofen, 5 mg kg⁻¹) and daily post-operative care (provided with hydrating gel, monitor food intake, and body weight). For multimodal device implantation, an

incision was made in the abdominal cavity and the device was implanted with the blue LED positioned towards the thoracic cavity and the green LED towards the abdominal cavity. After recovery from surgery, all animals received daily post-operative care and monitoring.

**Meal-pattern analysis.** To examine whether mice tolerate stomach device implantation, one group of mice was implanted with the device, whereas another group underwent a sham surgery where the abdomen was opened near the stomach, but it was not punctured nor implanted with a device. Two weeks after the surgeries, mice were placed in food-monitoring homecages (BioDAQ, v. 2.2). Feeding records were analyzed using BioDAQ Viewer (software v. 2.2.01). A feeding bout (≥0.01 g) was defined as a meal if ≥0.06 g of food was ingested and if it was separated from another meal by ≥5 min.

**Nodose ganglion injection.** With the mouse under anesthesia (isoflurane, 1–2% inhalation), a 1–2-cm-long skin incision was made from the left clavicle going upwards to the animal's jaw. The left vagus nerve was exposed by separating the platysma, sternohyoideus, and omohyoideus muscles using blunt dissection. After visualization of the ganglion, the virus (200 nl) was injected with a glass micropipette attached to a Nanoject II. The experimental group received AAV9-DIO-ChR2:tdTomato and control groups AAV9-DIO-tdTomato. The skin was then closed with interrupted stitches (6-0 silk suture). After the experiments, mice were anesthetized (Beuthanasia, 320 mg kg$^{-1}$ delivered intraperitoneally) and intracardially perfused with PBS, followed by 4% paraformaldehyde. Brains, nodose ganglion, and stomach tissues were then extracted, post fixed in 4% paraformaldehyde overnight, and cryoprotected in PBS containing 30% sucrose until the tissues sunk in the sucrose solution. Coronal cryostat sections for the brain, nodose, and stomach tissue were collected (30, 20, and 10 µm thick), directly mounted onto microscope slides, and coverslipped using DAPI Fluoromount-G mounting medium (SouthernBiotech). Two mice injected with AAV9-DIO-ChR2:tdTomato were excluded from behavioral analysis due to little (2–3 neurons infected per section) or no virus infection as determined by visualizing the fluorescent reporter, tdTomato.

**In situ hybridization.** Single-molecule fluorescence in situ hybridization was performed using an RNAscope Fluorescent Multiplex Kit (Advanced Cell Diagnostics). C1 and C3 DNA oligonucleotide probes were designed for *Calca* and *tdTomato*. Nodose sections were fixed in 4% paraformaldehyde for 15 min and then washed in 50, 70, 100, and 100% ethanol for 5 min each. Slides were dried for 5 min. Proteins were digested using protease solution (pretreatment solution 3) for 60–90 s. Immediately afterward, slides were washed twice in PBS. In parallel, C1 and C2 probes were heated in a 40 °C water bath for 10 min. Probes were applied to the slides, which were coverslipped and placed in a 40 °C humidified incubator for 3 h. Slides were rinsed twice in RNAscope wash buffer and then underwent the colorimetric reaction steps according to the standard kit protocol. After the final wash buffer, slides were immediately coverslipped using DAPI (4′,6-diamidino-2-phenylindole) Fluoromount-G mounting medium. Images were captured using a laser-scanning confocal (FV1200, Olympus) and epifluorescent (Eclipse E600, Nikon) microscopes.

**Measurement of µLED light spread.** Light intensity measurements (PM100D, Thorlabs) were acquired from fresh stomach tissue preparations ex vivo. Measurements were recorded after implanting the LED inside the stomach or after suturing the tether to the stomach surface with the LED directed towards the organ; the light sensor (S130C, Thorlabs) was encased in saran wrap and placed directly over the LED tether housing. We tested 2, 4, 6, 8, 10, and 12 W antenna power outputs. In a separate experiment, with LED not attached to a stomach, light dissipation was measured by placing the sensor 0, 0.5, and 1 cm away from the front- or backside of the LED housing.

**Fasting and refeeding experiments.** Mice were food restricted overnight (16 h) and refed the following morning. Food intake was manually measured 1, 2, and 3 h after refeeding. The same animals underwent multiple fasting–refeeding tests to examine different optogenetic stimulation parameters: no stimulation (RF antenna off), 10 Hz, and 20 Hz optogenetic stimulation (5 ms pulse width; RF power 4 W). Experiments were conducted 5 days apart.

**Real-time, place-preference (RTPP) assay.** Mice were placed in an RTPP box consisting of two chambers (20 cm × 18 cm) and a small transition area. Antennas were installed in both chambers, but only one chamber was connected to an RF generator to continuously deliver RF power (20 Hz, 5 ms pulse width, 4 W). The time spent in each chamber (20 min trial) was analyzed using video-tracking software (EthoVision XT 10, Noldus).

**Open-field test.** Mice were placed in the center of a 40 cm × 40 cm square open-field arena with non-transparent white Plexiglas. The total distance moved and time in the center (20 cm × 20 cm imaginary square), during the 10 min trial, were analyzed with video-tracking software with EthoVision. An RF antenna provided wireless power (20 Hz, 5 ms pulse width, 4 W) throughout the entire behavior box.

**Two-bottle flavor preference test.** Mice were accustomed to drinking water from two test tubes that replaced their normal water bottles. After acclimation, mice were water-deprived overnight, and the following morning received 30 min access to a novel 5% sucrose solution; immediately following sucrose consumption, mice received 4 h of optogenetic stimulation (20 Hz, 5 ms pulse width, 4 W). Three days later, mice were water-deprived overnight, and the following morning received 30 min access to separate test tubes containing either water or 5% sucrose solution. The intake of both solutions was measured and presented as a preference ratio (5% sucrose intake/total intake of sucrose and water solutions).

**Statistics.** Data were analyzed using Prism 5.0 (GraphPad software). Sample sizes were estimated based on prior experience and expected variability in feeding behavior[27]. We excluded an animal from data analysis if post hoc histological analysis showed no viral transduction as indicated by an absence of tdTomato fluorescence. For graphs comparing two experimental conditions, we used unpaired two-tailed Student's *t* test. We analyzed data sets (multiple treatments and time-points) with repeated-measures two-way analysis of variance tests (time repeated factor) and Tukey's post hoc tests. All data sets were conducted using Shapiro–Wilk normality test, and all passed the normality tests.

**Reporting summary.** Further information on research design is available in the Nature Research Reporting Summary linked to this article.

## Data availability
The main data supporting the results in this study are available within this article and its Supplementary information. Source data are provided with this paper.

## Code availability
The code for 3D reconstruction image is available from Github (https://github.com/parkgroup-tamu/3d_reconstruction) and also from Zenodo[40].

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

## Acknowledgements

This work was supported by grants from the interdisciplinary X-Grants Program, part of the President's Excellence Fund at Texas A&M University (S.P.), 2018 NARSARD Young Investigator Awards (S.P.) from Brain & Behavior Research Foundation, National Science Foundation Engineering Research Center for Precise Advanced Technologies and Health Systems for Underserved Populations PATHS-UP (EEC-1648451; S.P.), the University of Washington Diabetes Research Center (DK01747; C.C.) and National Institute of Health (DK12424; C.C.). S.P. would like to express thanks to Dr. Coté for general advice.

## Author contributions

W.K. designed a wireless optoelectronic platform, fabricated devices, tested devices, made wireless measurements, designed experiments, and generated figures. S.H. fabricated devices, tested devices, and conducted simulations of wireless platforms. M.G. performed a mechanical simulation of the device's structure. V.J. and C.M.S. implanted devices, tested mice in multiple cages, and did transmission antenna comparison. T.J.P. and R.D.P. provided resources and edited the manuscript. C.C. designed animal experiments, injected virus, implanted devices, conducted animal experiments, and analyzed behavioral data. C.C., S.P., and W.K. wrote the manuscript. S.P. oversaw all experiments and data analysis.

## Competing interests

The authors declare no competing interests.
