## [Peer Review File · Nature Communications]

Reviewers' Comments:

Reviewer #2:

Remarks to the Author:

This manuscript by Kim et al was an absolute pleasure to review. The work presented here is outstanding, informative and paradigm-shifting. The authors should be commended for their due diligence in testing various prototypes and design strategies, identifying weaknesses and optimizing them through various iterations. Finally, as proof of principle, they test the role of Calca vagal afferents in the stomach. I only have a couple comments listed below.

I'm a big believer in individual data points for bar graphs so SEM and numbers of subjects are more transparent so I suggest amending these.

For the fast-refeed studies, were the authors able to quantify the locomotor activity of the animals? Actually, now that I think of it it is probably far more informative and easier to extract this locomotor activity data from the RTPP and Open Field tests. Based on the representative traces it appears that Calca vagal stim doesn't seem to disrupt movement but I'm wondering if direct comparisons were made.

Also should mention that the Figures are beautiful and represent exactly what the text describes.

Reviewer #4:

Remarks to the Author:

The manuscript "Organ-specific, multimodal, wireless optoelectronics for high-throughput phenotyping of peripheral neural pathways" submitted by W.S. Kim and colleagues introduces a passive multimodal wireless optogenetic implantable device. This system implements wireless power transfer to power the device and enable battery free operation. With this system the authors demonstrate multimodal operation with a passive electronic circuitry with low power consumption, and long implantation time relevant for in-vivo experimentation. In addition, this piece of work also discusses a multiplexing approach that enables individual selection of a group of up to eight independent experimental cages without electromagnetic cross talk. The authors also include coupling between passive and active antennas as a strategy to enhance the RF coverage in the experimental enclosure. Using this technology, the authors demonstrate vagal nerve stimulation in the stomach of free-behaving mice under different behavioral paradigms. Although the work is well organized across the sections, and the technology development and in-vivo validation experiments are extremely impressive, the overall work seems to be disconnected and the novelty is not clear compared to other publications by this same group, as pointed out in the major comments described below. Thus, authors must provide convincing arguments and evidence on how these interesting technological improvements can facilitate neuroscience studies and how they are differentiated over past publications.

Major comments:

1. The multiplexing system is appealing and well executed but not employed in the in-vivo experiments. In the same way, the multimodal capability of the device is not used in the in-vivo experiment. With the lack of full integration of the technology presented here in the in-vivo demonstrations this work would be incremental to the concept previously published by the same group (Sensors 2020, 20, 3639) but adapted to a different stimulation organ.

Minor comments:

1. The authors state that previously published peripheral modulation optogenetic devices fail to

maintain their function after 8 days in the animal. However, the authors do not point out the reason for their failure and conceptually, how the device reported in the present paper improves the long-term operation.

2. The electrical circuits of the devices in Fig 1c and Fig 2f look different. And Fig 1c does not seem to have a reed switch. Are these two different designs?

3. Line 135 -140. The operational condition varies significantly for different optogenetic studies that require different stimulation dynamics. Author should include a few more details in the manuscript. For example, what are the stimulation conditions that lead to an increase of (~ 0.2 C).

4. Using a magnetic reed switch the authors achieve multimodal operation, selecting between two different optogenetics stimulation channels. In this regard, there is no clear description as per what are the stimulation parameters that this system can achieve when working in the multiplexed cage system.

5. Line 147 – 148. There are already commercial RF powering systems that control multiple cages through a single RF power generator, for simultaneous optogenetics stimulation in several cages at the same time. The author should correct this statement.

6. Line 144 – 163. The system reported here shows the capability to operate 8 cages simultaneously. However, there is no discussion on how this number of cages benefit certain neuroscience studies, especially those that require a specific group number of animals.

7. The multicage multiplexing approach is well executed. Although there is no crosstalk between the selected active antenna and the adjacent (detuned antennas), each cage has one passive antenna tuned at 15 MHz, which is close to the 13 MHz active antenna. Authors do not discuss this potential electromagnetic coupling.

8. Line 184 -186. It is confusing that extra heat dissipation in animal tissue results from the inefficient electromagnetic power transfer. The heat dissipation in animal tissue results from the device operation, which only depends on how much energy the device absorbs. If the wireless power transfer is inefficient, then the device will not operate as the result of lower magnetic field density in the cage, which means lower SAR.

9. Line 204 – 221. It is not clear what is the advantage of having 10 micro Watt for channel selection during long-term continuous behavior studies if the device is able to harvest more than that. What is the maximum power transfer efficiency this implantable device can achieve?. In this context, what is the power budget allocated to the electronics and how much optical power to the uLED.

Reviewer #5:

Remarks to the Author:

In this submission, Kim and coauthors reported an in vivo multimodal and wireless platform which enables the stable, long-term optogenetic stimulation of stomach vagal afferents. The authors showed that the developed flexible devices as well as the coil-antenna system for multiplexing powering enable the high-throughput experiments to investigate the behavioral and physiological phenotype with optogenetic modulation of peripheral nerve system in freely moving animals.

While some previous research showed the optoelectronic flexible devices targeting peripheral nerve system on the surface, it is still unique in this field to target the peripheral neurons 'inside' an organ which enables the stable, long-term use of the neural interfaces. While paying attention

this uniqueness, the reviewer also thinks this research is proposing the highly-practical, completed system for the experiments to investigate peripheral nerve system optogenetically which will make a positive effect in Neuroscience field. Moreover, the proposed devices have a large potential to be developed as electroceuticals in the future. Therefore, this reviewer recommends Nature Communications to publish this article after addressing some following issues, which would make the claim of the authors clearer and more durable:

The idea of using a pre-curved method and developing flexible/wireless devices using it is creative. Just addressing the following issues about the interfaces would be helpful to emphasize the advantages of the devices

- 1) At line 126 (when stating fig. 1d), the word 'strain' may be substituted in the 'stress'.
- 2) In supple 2a and b for PDMS layer, the same scale bar would be appropriate to show the greatness of the pre-curved insulation method.
- 3) Figure 1e, it is slightly confused what 0.72 and 2.87 means (radius of curvature maybe?). Can you explain what this graph means in the manuscript or caption more specifically? it would also be better to explain why both the case of 0.72 and 2.87 is worse than which of 1.15

The proposed cage system seems also practical. Can you describe how many numbers of cages in the maximum are possible to operate at the same time? Of course, the duty rate and stimulation time (~5ms) of optogenetic stimulation as well as the switching time of the system should be more important, but providing the estimated number with some rationale would be beneficial for the neuroscientists who will use your system in the future.

To the readers who are not experts about wireless communication like me, the numbers about the intensity of H field in each figure are a little bit confused. For example, the maximum value of H field is $2.5 \sim 3$ A/m in Fig 2c and Supple fig. 11f, while which is 20 A/m in Fig 2e and Supple Fig.10a. Can you explain why, or clarify the reason in the manuscript if needed?

Also, it is hard to understand why only the H fields in supple fig 11a shows the donut-shape while other figures show the filled-rectangular-shaped field. Moreover, if I understand right, position C in Supple Fig. 11d should show less device-received power than other positions due to this donut-shaped H field. Can you explain about it?

The data in Supple Fig. 6 about the change in temperature by optical density of 10 mW/mm^2 is highly informative. Is it also possible for you to provide the estimated number of temperature changes considering all the systems including cages and devices in the real experiments? For example, you can say the 20 A/m H field by the caging system would make $XX \text{ mW/mm}^2$ of the optical density, and it will make the $XX^\circ\text{C}$ change in temperature.

In Fig 3a, the value is maybe about the light intensity leaked out from the stomach (mainly backside scattering). If I'm right, it is hard to understand how the measured intensity outside the organ can be related to the ChR2 activation. In the same context, it may be better to show the relative value instead of absolute light intensity? (or indicate the light intensity from the device together). Please explain your rationale about it.

If possible, it would be better to show the simulation data about heat dissipation from your device 'in the stomach'. It would be useful to emphasize the thermal safeness of your device.

Can you explain how to distinguish the effect of less hunger and anxiety in your experiments? You explained the suppressed appetite by activation of Calca+ stomach vagal afferents also increased the anxiety level of the mice. However, is there no possibility just increased anxiety level of mice makes them less hungry?

Lastly, it would be striking if you can show how long your interfaces can remain stable in the

stomach of freely moving animals. I think it is one of the strongest points of your research.

Reviewer #2 (Remarks to the Author):

Comments:

This manuscript by Kim et al was an absolute pleasure to review. The work presented here is outstanding, informative and paradigm-shifting. The authors should be commended for their due diligence in testing various prototypes and design strategies, identifying weaknesses and optimizing them through various iterations. Finally, as proof of principle, they test the role of Calca vagal afferents in the stomach. I only have a couple comments listed below.

1. I'm a big believer in individual data points for bar graphs so SEM and numbers of subjects are more transparent so I suggest amending these. For the fast-refeed studies, were the authors able to quantify the locomotor activity of the animals? Actually, now that I think of it, it is probably far more informative and easier to extract this locomotor activity data from the RTPP and Open Field tests. Based on the representative traces it appears that Calca vagal stim doesn't seem to disrupt movement but I'm wondering if direct comparisons were made. Also should mention that the Figures are beautiful and represent exactly what the text describes.

Response: Aside from the supplemental video, we did not video record food intake behavior. However, as suggested, we can extrapolate locomotor activity from RTPP and Open Field tests. We have included this data in **supplemental figure 18** and added the following text in the last paragraph in Result section: **"We analyzed the locomotor activity from open field and place preference tests, which revealed decreased locomotion during optogenetic activation of Calca+ gastric vagal afferent (Supplementary Fig. 18) ; presumably, the mice feel aversion and have decreased motivation. In addition,"**. The bar graphs have also been updated to include individual data points, thank you for the suggestion.

Reviewer #4 (Remarks to the Author):

Comments:

The manuscript “Organ-specific, multimodal, wireless optoelectronics for high-throughput phenotyping of peripheral neural pathways” submitted by W.S. Kim and colleagues introduces a passive multimodal wireless optogenetic implantable device. This system implements wireless power transfer to power the device and enable battery free operation. With this system the authors demonstrate multimodal operation with a passive electronic circuitry with low power consumption, and long implantation time relevant for in-vivo experimentation. In addition, this piece of work also discusses a multiplexing approach that enables individual selection of a group of up to eight independent experimental cages without electromagnetic cross talk. The authors also include coupling between passive and active antennas as a strategy to enhance the RF coverage in the experimental enclosure. Using this technology, the authors demonstrate vagal nerve stimulation in the stomach of free-behaving mice under different behavioral paradigms. Although the work is well organized across the sections, and the technology development and in-vivo validation experiments are extremely impressive, the overall work seems to be disconnected and the novelty is not clear compared to other publications by this same group, as pointed out in the major comments described below. Thus, authors must provide convincing arguments and evidence on how these interesting technological improvements can facilitate neuroscience studies and how they are differentiated over past publications.

1. (Major) The multiplexing system is appealing and well executed but not employed in the in-vivo experiments. In the same way, the multimodal capability of the device is not used in the in-vivo experiment. With the lack of full integration of the technology presented here in the in-vivo demonstrations this work would be incremental to the concept previously published by the same group (*Sensors* 2020, 20, 3639) but adapted to a different stimulation organ.

Response: All feeding behavior results utilized the multiplexing system to efficiently examine multiple hours of food intake under various stimulation parameters (with recovery days in between experimental sessions). This enabled us to complete the feeding experiments and then conduct other behavioral measures in the same animals while the device was functional. Because the device is implanted into the stomach and not readily visible without scruffing the animal, we demonstrated the multiplexing system with the devices *in vitro* (video 1) and implanted on the skull (video 2).

The multimodal capability is a proof concept for this paper. Implementation *in vivo* requires validation of additional viruses to deliver other light-activated opsins and a physiologically-relevant experimental question to address, which is beyond the scope of our current report. Efforts to address relevant questions are ongoing.

In addition, the concept and functionality of the multi-channel device in this paper is clearly distinct from the previously published approach in *Sensors*. Previously, we suggested two functionalities: 1) the first switching mechanism should be actuated by a magnet, and 2) another mechanism, that channel 1 could lit on only constantly. So, mechanism 1) had not selected the channel by RF pulse signal, and mechanism 2) had limited capabilities to use two-channels for using the stimulation signal. It means, both functionalities had limited performance, especially to use dual-channel with neuron activation. In this paper, however, we highlighted that both channels enable activation (pulse emission) and deactivation (constant emission) of neurons, independently, and channel selection induced by simple RF pulse with magnet-free. Previously published concept represented detailed explanations (e.g. The signal flows of circuits in figure S2, S4 of Kim et al. *Sensors*). Supplementary Figure 14 shows a circuit diagram of the proposed dual-channel optoelectronic device. The following circuit diagrams provide detailed explanations about channel operation.

Proposed dual-channel optoelectronic device. Signal-flows from off mode to LED ch1, and from LED ch1 to LED ch2.

Figure S4. (a) Circuit diagrams of a dual channel device that offers an advanced magnet-free operational mode-inhibition and stimulation of neural activity. Here, a dual channel device automatically activates a channel in response to signals from a remotely located wireless TX system. $R1 = 20k\Omega$ and a reed switch is denoted by S1. (b)-(d) Signal-flows from a power supply to Ch1 LED for Ch1 activation. (e)-(j) Signal-flows during switching from Ch1 to Ch2.

1. (Minor) The authors state that previously published peripheral modulation optogenetic devices fail to maintain their function after 8 days in the animal. However, the authors do not point out the reason for their failure and conceptually, how the device reported in the present paper improves the long-term operation.

Response: Previously published studies about the peripheral optogenetic devices did not test device durability nor clearly mention why failure occurred after one week of *in vivo* operation. Thus, we are not able to directly infer the potential reasons for failure. For our device, we outline in the Results the testing and alterations that led to increased durability, such as constructing the tether in a U shape and sandwiching the components in between two copper-polyamide bilayers. We compared these design alterations to a standard construction, which was used by other previously published approaches, and found significant improvement. While the U shape is unique to our affixation approach that increases organ specificity, sandwiching components between copper-polyamide bilayers are likely to also increase durability of other devices that are affixed to the surface of an organ. Additionally, it is noteworthy that our energy harvester is 3 times smaller than other devices (e.g. ~ 3 cm shown in figure 1b of Mickle et al.). Our small harvester is therefore less likely to undergo strains and stresses in the abdomen during animal movement compared to other published devices.

2. (Minor) The electrical circuits of the devices in Fig 1c and Fig 2f look different. And Fig 1c does not seem to have a reed switch. Are these two different designs?

Response: Yes, both are different designs. Fig 1c is for one channel device without a reed switch, and Fig 2f represents a dual-channel device, including a reed switch.

3. (Minor) Line 135 -140. The operational condition varies significantly for different optogenetic studies that require different stimulation dynamics. Author should include a few more details in the manuscript. For example, what are the stimulation conditions that lead to an increase of (~0.2 C).

Response: We agree that details in operational conditions should be provided. We made modifications to sentences in the last paragraph of Results 1st subheading section: “Thermal assessment of the wireless optogenetic implant demonstrated minimal temperature increases (~0.2 °C; Supplementary Fig. 6) during typical operating conditions (10- and 20-Hz with 5-ms light pulse; 10 % and 5 % duty cycles).”

4. (Minor) Using a magnetic reed switch the authors achieve multimodal operation, selecting between two different optogenetics stimulation channels. In this regard, there is no clear description as per what are the stimulation parameters that this system can achieve when working in the multiplexed cage system.

Response: We agree that such descriptions should be included. We added the following sentences in Results 3rd subheading section: “Time slots allocated for each cage and threshold pulse for channel selection are tunable, suggesting many scenarios of multiplexing and multimodal operation. For example, we set threshold pulse for activation/deactivation of channels to 100 ms and allocate 250 ms for each cage. This provides enough time for an implant in each cage to switch its channel (from Ch1 to Ch2 or vice versa). It requires only two seconds (8×0.25 sec) for switching operation of implants in cages. Next, the TX system can adjust time slots, depending on stimulation conditions.”

5. (Minor) Line 147 – 148. There are already commercial RF powering systems that control multiple cages through a single RF power generator, for simultaneous optogenetics stimulation in several cages at the same time. The author should correct this statement.

Response: Yes, we agree that the current commercial system supports the multiplexing function. However, the commercial system occurs electromagnetic interference with next to each other. Thus, we mentioned functionality and limitation of this system in the 1st paragraph of Results 2nd subheading section: “Multiple RF power generators can be used, but they must be operated at least 1 m apart from each other to avoid electromagnetic interface (Supplementary Fig. 8).” In contrast, our telemetry system requires only spaces (0.6×1 m) to accommodate 8 homecages.

6. (Minor) Line 144 – 163. The system reported here shows the capability to operate 8 cages simultaneously. However, there is no discussion on how this number of cages benefit certain neuroscience studies, especially those that require a specific group number of animals.

Response: We agree it should be discussed. We added the following sentences in the 2nd paragraph of Results 2nd subheading section: “Therefore, the limiting factors for the number of cages that can be operated simultaneously with a single RF-power generator are the stimulation frequency and duration of light pulses. With the proposed arrangement (Fig. 2a), we conducted experiments within 8 cages simultaneously using 20-Hz and 5-ms pulse duration stimulation parameters. This extends the high-throughput utilizing optogenetics to do the experiments at least 8-mice (supposed a mouse in a cage), a typical group size, at that same time. For example, measurements of food intake require 4 hours for each animal in the group. To complete analysis of feeding behavior for two groups of animals (experimental and control), each of which has 8

animals, it only takes 8 hours while approaches using existing wireless TX system, a single power source coupled with a single cage, demand 64 hours (8 × 8 hours). This makes it less ideal for longitudinal experiments, in particular those required for most obesity experimental designs, where a device needs to be chronically implanted for >2 months. Also, through the modification of the multiplexer board and controller, simultaneous activation of 16 cages at 10-Hz or 32 cages at 5-Hz frequency is possible when using a 5-ms light pulse duration.”

7. (Minor) The multicage multiplexing approach is well executed. Although there is no crosstalk between the selected active antenna and the adjacent (detuned antennas), each cage has one passive antenna tuned at 15 MHz, which is close to the 13 MHz active antenna. Authors do not discuss this potential electromagnetic coupling.

Response: We agree that clarifications are needed. The detuning switch is applied to not only the top coil antenna but also the passive antenna. Device operation relies on magnetic resonant couplings between a coiled antenna and an implant. Magnetic resonant transmission system offers narrow bandwidth due to its high quality factor (> 200), suggesting that power transmission occurs only when a resonant frequency of the TX system is matched to that of an implant, 13.56MHz. Here, a resonant frequency of a coil antenna tuned at 15 MHz is shifted toward 13.56MHz when activated (resonated). However, the resonant frequency of the coil antenna is shifted toward high frequency range above 20 MHz when deactivated by RF and Decoupling multiplexer. At such high level of deviation from 13.56MHz, couplings are negligible (below 30 dB; 0.01%)

Accordingly, we modified the text so it is clear that the detuning switch is applied to both the top and bottom antenna (antenna set) in the 2nd paragraph of Results 2nd subheading section: “The wireless telemetry system consists of an RF- power generator, controller, RF multiplexer, decoupling multiplexer, and an antenna set for each of the 8 cages (Fig. 2a); each antenna set is made of a pair of top and bottom coil structure...For example, when the controller selects antenna set 6, the RF multiplexer tunes antenna set 6 to 13.56 MHz and Decoupling multiplexer detunes the other antenna sets to 100 MHz (Fig. 2b)... Therefore, the other 7 antenna sets do not cause interference even when directly adjacent to the actuating 13.56 MHz antenna...”

8. (Minor) Line 184 -186. It is confusing that extra heat dissipation in animal tissue results from the inefficient electromagnetic power transfer. The heat dissipation in animal tissue results from the device operation, which only depends on how much energy the device absorbs. If the wireless power transfer is inefficient, then the device will not operate as the result of lower magnetic field density in the cage, which means lower SAR.

Response: Light sources could generate heat during operation. However, what we intend to convey is that RF signals transmitted from a remotely located wireless power TX system can increase heat by directly interacting with animal tissue, thus more heat dissipation with great RF power. This is why IEEE or FCC suggested guidelines in regard to limits to maximum exposure to RF waves. We modified the following sentence to make this clear in the 3rd paragraph of Results 2nd subheading section: “Previously, these limitations were circumvented by increasing RF power, but this results in undesired RF energy to animal tissues and increased heat generation.”

9. (Minor) Line 204 – 221. It is not clear what is the advantage of having 10 micro Watt for channel selection during long-term continuous behavior studies if the device is able to harvest more than that. What is the maximum power transfer efficiency this implantable device can achieve? In this context, what is the power budget allocated to the electronics and how much optical power to the uLED.

Response: Other devices with a microcontroller for multichannel manipulation requires 1000-folds (10 ~ 30 mW) greater amount of power than ours (10 µW). This suggests that the TX system must deliver power above the threshold (>10 mW) to an implant for channel selection, and it

renders a wireless device energy hungry. We can boost TX powers, but it could lead to tissue damages caused by the absorption of RF waves into biological tissues. In contrast, the proposed wireless telemetry system can enable robust activation of an implant and channel switching throughout the volume of the cage due to extremely low power consumption of the implant.

Transmission efficiency is 60 %, and each implant can harvest up to 10 mW (electrical power), corresponding to an optical power density of 200 mW/mm². (optical power intensity required for light-sensitive opsins is 10 mW/mm².)

Reviewer #5 (Remarks to the Author):

Comments:

In this submission, Kim and coauthors reported an in vivo multimodal and wireless platform which enables the stable, long-term optogenetic stimulation of stomach vagal afferents. The authors showed that the developed flexible devices as well as the coil-antenna system for multiplexing powering enable the high-throughput experiments to investigate the behavioral and physiological phenotype with optogenetic modulation of peripheral nerve system in freely moving animals.

While some previous research showed the optoelectronic flexible devices targeting peripheral nerve system on the surface, it is still unique in this field to target the peripheral neurons 'inside' an organ which enables the stable, long-term use of the neural interfaces. While paying attention this uniqueness, the reviewer also thinks this research is proposing the highly-practical, completed system for the experiments to investigate peripheral nerve system optogenetically which will make a positive effect in Neuroscience field. Moreover, the proposed devices have a large potential to be developed as electroceuticals in the future. Therefore, this reviewer recommends Nature Communications to publish this article after addressing some following issues, which would make the claim of the authors clearer and more durable:

1. The idea of using a pre-curved method and developing flexible/wireless devices using it is creative. Just addressing the following issues about the interfaces would be helpful to emphasize the advantages of the devices

1) At line 126 (when stating fig. 1d), the word 'strain' may be substituted in the 'stress'.

2) In suppl 2a and b for PDMS layer, the same scale bar would be appropriate to show the greatness of the pre-curved insulation method.

3) Figure 1e, it is slightly confused what 0.72 and 2.87 means (radius of curvature maybe?). Can you explain what this graph means in the manuscript or caption more specifically? it would also be better to explain why both the case of 0.72 and 2.87 is worse than which of 1.15

Response: 1) We agree with the reviewer's suggestion and modified the word to 'strain'.

2) We added more figures of the same scale bar level in Supplementary Figure 2, thank you for the suggestion.

3) The calculation method and equation for the curvature is denoted in Supplementary Figure 3. A 2.87 curvature is too similar to the flat structure and thus exhibits poor durability; in contrast, 0.72 is too much curved not to keep μ LED attached to the pad correctly; this bending ratio makes μ LED easily detached from the pattern by the higher stress. A brief explanation of the number is added in Figure 1e caption: "The legend numbers represent the curvature of the device as the length of the arc for a radius of 2 mm (see Supplemental Fig. 3 for equation); smaller numbers represent sharper U-shape angle." And add the rationale in the 3rd paragraph of Results 1st subheading section: "Although there was improved durability with pre-curved structures that had a radius of 2.87 and 0.72 mm, they were not as durable as 1.15 mm, likely because 2.87 mm is too similar to the flat structure, whereas the sharp angle with a radius of 0.72 mm interferes with μ LED contact with the pad." Also, we modified figure 1d (left) and added figures in Supplementary Figure 2 (right).

2. The proposed cage system seems also practical. Can you describe how many numbers of cages in the maximum are possible to operate at the same time? Of course, the duty rate and stimulation time (~5ms) of optogenetic stimulation as well as the switching time of the system should be more important, but providing the estimated number with some rationale would be beneficial for the neuroscientists who will use your system in the future.

Response: The point is well-taken. We added the following sentences in the 2nd paragraph of Results 2nd subheading section: “Therefore, the limiting factors for the number of cages that can be operated simultaneously with a single RF-power generator are the stimulation frequency and duration of light pulses. With the proposed arrangement (Fig. 2a), we conducted experiments within 8 cages simultaneously using 20-Hz and 5-ms pulse duration stimulation parameters. This extends the high-throughput utilizing optogenetics to do the experiments at least 8-mice (supposed a mouse in a cage), a typical group size, at that same time. For example, measurements of food intake require 4 hours for each animal in the group. To complete analysis of feeding behavior for two groups of animals (experimental and control), each of which has 8 animals, it only takes 8 hours while approaches using existing wireless TX system, a single power source coupled with a single cage, demand 64 hours (8×8 hours). This makes it less ideal for longitudinal experiments, in particular those required for most obesity experimental designs, where a device needs to be chronically implanted for >2 months. Also, through the modification of the multiplexer board and controller, simultaneous activation of 16 cages at 10-Hz or 32 cages at 5-Hz frequency is possible when using a 5-ms light pulse duration.”

3. To the readers who are not experts about wireless communication like me, the numbers about the intensity of H field in each figure are a little bit confused. For example, the maximum value of H field is 2.5~3 A/m in Fig 2c and Supple fig. 11f, while which is 20 A/m in Fig 2e and Supple Fig.10a. Can you explain why, or clarify the reason in the manuscript if needed?

Response: We agree that clarifications are needed. We previously considered the beauty of the color or shape for the readers. We set all scale bar levels, 20 A/m max, and re-simulated with Cu strip and wire structure, and modified Fig 2c and Supplementary Figure 9b & 11f.

4. Also, it is hard to understand why only the H fields in supple fig 11a shows the donut-shape while other figures show the filled-rectangular-shaped field. Moreover, if I understand right, position C in Supple Fig. 11d should show less device-received power than other positions due to this donut-shaped H field. Can you explain about it?

(4) Our response: The reviewer's point is well-taken. A result such as the donut-shaped issued by wrong radiation box size in the simulation. The radiation box is one of the key factors for simulating the H-fields since it limits the region of the electromagnetic field when the simulation run and calculate values. To be clear, the following images 1), 2), and 3) shows how different size of radiation boxes effect on the simulation result; it may help to understand this intuitively. We set it very narrow (such as 1), the left image below) to add 4-layers in one picture; this made the non-sense result in supplementary fig 11. So, we re-simulated all with the correct range such as 3), right image, and all simulations are **updated in supplementary fig 11**.

5. The data in Supple Fig. 6 about the change in temperature by optical density of 10mW/mm² is highly informative. Is it also possible for you to provide the estimated number of temperature changes considering all the systems including cages and devices in the real experiments? For example, you can say the 20 A/m H field by the caging system would make XX mW/mm² of the optical density, and it will make the XX°C change in temperature.
7. If possible, it would be better to show the simulation data about heat dissipation from your device 'in the stomach'. It would be useful to emphasize the thermal safeness of your device.

Response: The reviewer's point is well-taken. To reduce the gap and reflect the reviewer's suggestion, we measured the temperature variations in various cases *in vitro* as same as circumstances in animal experiments. For example, we set the TX power level as same as not only the HFSS simulation parameters, but also *in vivo* experiment condition, 4 W, and place the device inside of the cage that usually device implanted mouse stay in real. We conducted these measurements with the pre-curved device in dry, wet, and in the 10 % PBS solution that is assuming it as the mouse body. Wireless measurements of optical intensity using IR camera at various duty cycles revealed that there is no detectable change in temperature during operation (see the below plots). We added images of the experimental setup and measurements of optical intensity at duty cycles (5 and 10 %) in Supplementary Figure 6.

For clarification of the optical density in simulation result, we added a guideline in the plot, supplementary fig 11d. The guideline value, 0.58 mW is can be converted to the optical density >10 mW/mm² of proposed device. The conversion efficiency of the LEDs is 30%. Electrical power of 0.58 mW corresponds to optical intensity of 10.12 mW/mm² (0.58mW, electrical power, x 0.3, conversion efficiency, / 0.0172 mm², a junction area of LED). We modified a plot in Supplementary fig11d and caption: "Dotted lines indicate the threshold electrical power (0.58 mW) required for the activation of light-sensitive opsins; 0.58 mW electrical power corresponds to an optical power of 10.12 mW/mm²."

Regarding heat dissipation simulation, this may be out of scope since it requires developing a simulation program. Instead of the simulation, we conducted measurements of temperature variations of the device in a real experimental condition. We added these results as plots in Supplementary Figure 6c-e.

Supplementary Figure 6. Measurements of temperature changes as a function of operating duty cycles in wet (a) and dry condition (b) using DC power inputs for fixing the optical density of 10 mW/mm². Pictures of an experimental setup for wireless measurements of heat dissipation using IR camera (c; top). Here, TX power is set to 4 W. The three bottom images show a device mounted on sealed bag of saline solution (10 % PBS), immersed in saline solution, and itself in a cage, respectively. Plots of optical intensity as a function of time at duty cycles (d; 5 %) and (e; 10 %) in three different conditions; wet, dry, and PBS bag, respectively. The following provides a guideline for comparison of parameters in experimental settings with those in simulation settings. Magnetic field intensity of 20 A/m in simulation results corresponds to electrical power of 4 W or optical intensity of 10 W/mm² in experimental settings.

6. In Fig 3a, the value is maybe about the light intensity leaked out from the stomach (mainly backside scattering). If I'm right, it is hard to understand how the measured intensity outside the organ can be related to the ChR2 activation. In the same context, it may be better to show the relative value instead of absolute light intensity? (or indicate the light intensity from the device together). Please explain your rationale about it.

Response: The interpretation of Fig 3a is correct; our goal was to determine the light intensity leaking from outside the stomach. The values are reported as absolute light intensity (mW/mm^2) because these are the values used to approximate the amount of opsin activation that may occur in surrounding tissues when the LED is not implanted inside the stomach (see dashed lines indicating 10% and 50% opsin channel activation, based on published data).

8. Can you explain how to distinguish the effect of less hunger and anxiety in your experiments? You explained the suppressed appetite by activation of Calca+ stomach vagal afferents also increased the anxiety level of the mice. However, is there no possibility just increased anxiety level of mice makes them less hungry?

Response: This is a good point, although it is difficult to distinguish the effects of hunger/satiety and anxiety. Under physiological conditions, anxiety is observed immediately following food consumption and satiation. The prevailing hypothesis is that hunger decreases anxiety to increase risk taking and facilitate foraging, and the inverse occurs after a meal when foraging is not immediately beneficial. Additional experiments, such as conditioned taste aversion, are useful for interpreting the effects of activating neurons on food intake and anxiety. In this case, activating Calca+ stomach vagal afferents induced a conditioned taste aversion, suggesting that this neuronal population might be involved in mediating aversive responses (e.g. in response to food poisoning). This is an active area of research that our labs are pursuing.

9. Lastly, it would be striking if you can show how long your interfaces can remain stable in the stomach of freely moving animals. I think it is one of the strongest points of your research.

Response: The durability data is included in Fig 1g.

Reviewers' Comments:

Reviewer #2:

Remarks to the Author:

The authors have addressed my two concerns.

Reviewer #4:

Remarks to the Author:

The authors have provided thorough responses to all reviewer inputs. I believe that the revised version is suitable for publication in its current form.

Reviewer #5:

Remarks to the Author:

This reviewer thinks the authors addressed all the comments successfully, so recommend for Journal to publish the manuscript.

Minor comments about statistical analysis:

1. The following information for every statistical analysis (Fig. 1g, 3a, 3b, 4b, 4c, 4e, 4h, 4j, Supple Fig. 15, 16, 18) need to be included.
 - 1) normality test
 - 2) covariance test
 - 3) statistical methods
2. Statistical importance based on the p-values (e.g. *, **, ***) needs to be included for Fig. 1g, 3a, 3b, Supple Fig. 15)

Reviewer #2 (Remarks to the Author):

Comments:

The authors have addressed my two concerns.

Reviewer #4 (Remarks to the Author):

Comments:

The authors have provided thorough responses to all reviewer inputs. I believe that the revised version is suitable for publication in its current form.

Reviewer #5 (Remarks to the Author):

Comments:

This reviewer thinks the authors addressed all the comments successfully, so recommend for Journal to publish the manuscript.

Minor comments about statistical analysis:

1. The following information for every statistical analysis (Fig. 1g, 3a, 3b, 4b, 4c, 4e, 4h, 4j, Supple Fig. 15, 16, 18) need to be included.

- 1) normality test
- 2) covariance test
- 3) statistical methods

2. Statistical importance based on the p-values (e.g. *, **, ***) needs to be included for Fig. 1g, 3a, 3b, Supple Fig. 15)

Response: The reviewer's point is well-taken. We modified and added the following sentences in the Methods section under Statistics: "We analyzed data sets (multiple treatments and time-points) with repeated-measures two-way ANOVA tests (time repeated factor) and Tukey's post-hoc tests. All data sets are conducted Shapiro-Wilk normality test; and all pass normality." Also, we added the following sentences in the caption of Figure 1, 3, and 4: "Statistical comparison was made using two-tailed t-test; *** p < 0.001.", "Statistical comparisons were made using two-way repeated-measures ANOVA, Tukey's post hoc; *** p < 0.001.", "Statistical comparisons were made using two-way repeated-measures ANOVA, Tukey's post hoc, except for h and j, which were two-tailed t-tests; ** p < 0.01, *** p < 0.001." Lastly, we modified the asterisks and captions in Supplementary figure 15, 16, and 18.

Supplementary Figure 15. Light intensity measurements during varying RF wireless powering ($p < 0.001$) of the gastric optogenetic device ($n = 5$) and varying distances ($p < 0.001$) from the LED. Measurements were taken from the front side (**a**), back side (**b**), and lateral side of the LED (**c**). Bar graphs are mean \pm SEM. Statistical comparisons were made two-way repeated-measures ANOVA; *** $p < 0.001$.

Supplementary Figure 16. Abdominal activation of Calca+ vagal afferent fibers. **(a)** Picture showing wirelessly powered LED device; two of these devices were inserted into the abdomen of Calca-Cre transgenic mice with left nodose ganglion injection of AAV9-DIO-ChR2:tdTomato or AAV9-DIO-tdTomato control virus; scale bar 5 mm. **(b)** top, frequency dependent suppression of food intake during ChR2 activation of vagal afferents ($n = 8$), bottom, no appetite suppression in tdTomato control group ($n = 4$) ($p = 0.80$). **(c)** Percent reduction of food intake (compared to RF off) during 10 and 20 Hz stimulation of Calca+ vagal afferent endings using the stomach LED implant or non attached LEDs (ChR2, $n = 8$ per group) (interaction, $p = 0.01$). Bar graphs are mean \pm SEM. Statistical comparisons were made two-way repeated-measures ANOVA, Tukey's post hoc; * $p < 0.05$; ** $p < 0.01$; *** $p < 0.001$.

Supplementary Figure 18. Locomotor activity comparison in the assays for open-field **(a)**, and RTPP **(b)**, respectively. Both conducted for 30 minutes and $n = 7$ per each group. Bar graphs are mean \pm SEM. Statistical comparison was made using two-tailed t-test; *** $p < 0.001$.